# Identifying Features of a System of Practice to Inform a Contemporary Competency Framework for Paramedics in Canada

**DOI:** 10.3390/healthcare12090946

**Published:** 2024-05-05

**Authors:** Alan M. Batt, Meghan Lysko, Jennifer L. Bolster, Pierre Poirier, Derek Cassista, Michael Austin, Cheryl Cameron, Elizabeth A. Donnelly, Becky Donelon, Noël Dunn, William Johnston, Chelsea Lanos, Tyne M. Lunn, Paige Mason, Sean Teed, Charlene Vacon, Walter Tavares

**Affiliations:** 1Faculty of Health Sciences, Queen’s University, 99 University Avenue, Kingston, ON K7L 3N6, Canada; 2Department of Paramedicine, Monash University, Building H, Peninsula Campus, 47–49 Moorooduc Hwy, Frankston, VIC 3199, Australia; jennifer.bolster@monash.edu (J.L.B.); or cheryl@virtualhospice.ca (C.C.); chelsea.lanos@monash.edu (C.L.); tyne.lunn@monash.edu (T.M.L.); 3Oxford County Paramedic Services, 377 Mill Street, Woodstock, ON N4S 7V6, Canada; meghan.lysko@gmail.com; 4BC Emergency Health Services, Clinical Governance and Professional Practice, 2955 Virtual Way, Vancouver, BC V5M 4X3, Canada; 5Paramedic Association of Canada, 201-4 Florence Street., Ottawa, ON K2P 0W7, Canada; pierre.poirier@paramedic.ca (P.P.); derek.cassista@paramedic.ca (D.C.); 6Ottawa Paramedic Service, 2465 Don Reid Drive, Ottawa, ON K1H 1E2, Canada; johnstcw@gmail.com (W.J.);; 7Department of Emergency Medicine, The Ottawa Hospital, University of Ottawa, 2475 Don Reid Drive, Ottawa, ON K1H 1E2, Canada; maustin@toh.ca; 8Canadian Virtual Hospice, One Morley Avenue, Winnipeg, MB R3L 2P4, Canada; 9School of Social Work, University of Windsor, 167 Ferry Street, Windsor, Ontario, ON N9A 0C5, Canada; donnelly@uwindsor.ca; 10Health Sciences Division, Justice Institute of British Columbia, 715 McBride Boulevard, New Westminster, BC V3L 5T4, Canada; bdonelon@jibc.ca; 11Saskatchewan Health Authority, 1350 Albert Street, Regina, SK S4R 2R7, Canada; noel.dunn@saskhealthauthority.ca; 12School of Interdisciplinary Studies, Royal Roads University, 2005 Sooke Road, Victoria, BC V9B 5Y2, Canada; 13School of Paramedicine, Medavie HealthEd, 50 Eileen Stubbs Avenue, Unit 154, Dartmouth, NS B3B 0M7, Canada; sean.teed@medaviehealthed.com; 14Regional Paramedic Program for Eastern Ontario, The Ottawa Hospital, 2475 Don Reid Drive, Ottawa, ON K1H 1E2, Canada; lcvacon@gmail.com; 15Department of Health and Society & Wilson Centre for Health Professions Education Research, University of Toronto, 1265 Military Trail, Toronto, ON M1C1A4, Canada; walter.tavares@utoronto.ca; 16York Region Paramedic Services, 80 Bales Drive East, East Gwillimbury, ON L0G 1V, Canada

**Keywords:** systems thinking, paramedicine, Canada, paramedic practice, competency

## Abstract

Introduction: Paramedic practice is highly variable, occurs in diverse contexts, and involves the assessment and management of a range of presentations of varying acuity across the lifespan. As a result, attempts to define paramedic practice have been challenging and incomplete. This has led to inaccurate or under-representations of practice that can ultimately affect education, assessment, and the delivery of care. In this study, we outline our efforts to better identify, explore, and represent professional practice when developing a national competency framework for paramedics in Canada. Methods: We used a systems-thinking approach to identify the settings, contexts, features, and influences on paramedic practice in Canada. This approach makes use of the role and influence of system features at the microsystem, mesosystem, exosystem, macrosystem, supra-macrosystem, and chronosystem levels in ways that can provide new insights. We used methods such as rich pictures, diagramming, and systems mapping to explore relationships between these contexts and features. Findings: When we examine the system of practice in paramedicine, multiple layers become evident and within them we start to see details of features that ought to be considered in any future competency development work. Our exploration of the system highlights that paramedic practice considers the person receiving care, caregivers, and paramedics. It involves collaboration within co-located and dispersed teams that are composed of other health and social care professionals, public safety personnel, and others. Practice is enacted across varying geographical, cultural, social, and technical contexts and is subject to multiple levels of policy, regulatory, and legislative influence. Conclusion: Using a systems-thinking approach, we developed a detailed systems map of paramedic practice in Canada. This map can be used to inform the initial stages of a more representative, comprehensive, and contemporary national competency framework for paramedics in Canada.

## 1. Introduction

Defining or describing the competencies necessary for paramedic practice in Canada is challenging given the diverse range of clinical events and variable contexts in which they practice. This requires a broad set of competencies that can be difficult to define due to the complexity, unpredictable nature, and broad range of clinical events and demands that then become inherently associated with paramedic practice. Clinical events may include a range of acute or chronic medical- and trauma-related concerns, comorbidities, and complex social conditions across the life span. Much like the utilization of emergency departments [1], the public has expanded its use of paramedic systems, without much restriction on when or where these events occur (e.g., industrial sites, homes, clinics, workplaces, remote communities, wilderness) [2,3,4,5,6], differentiating paramedics from nearly every other health professional. Consequently, scopes of practice and models of care are continuously shifting and expanding. For instance, emerging community-based models of care involve paramedics functioning beyond traditional emergency response, such as follow-up services for emergency calls, prevention programs, and active engagement in home support [7,8,9,10]. This evolving diversity of practice presents a challenge when attempts are made to describe it.

This challenge has resulted in a disconnect between the experiences of paramedics and the competencies and, ultimately, educational practices that guide them. Despite paramedics caring for people from a variety of environmental, social, and cultural contexts, these and likely other factors have traditionally been poorly addressed. For example, while older adults account for approximately 50% of paramedic patient encounters [11], there is a noticeable lack of attention on this patient population in paramedic curricula across Canada. Paramedics regularly care for people with chronic and undifferentiated disease presentations, yet these people are often not part of the interdisciplinary primary care workforce [12,13]. Paramedic education—and the competency frameworks that guide it [14]—remains largely focused on acute exacerbations and an isolated, event-based model of care. This pattern of misalignment of paramedic experiences with paramedic education is particularly noticeable when we look for competencies intended to guide the care of marginalized and under-served populations, such as those who are victims of human trafficking and modern-day slavery, or for those experiencing mental health issues, substance use dependency, and homelessness [15,16,17].

Regional variations present an additional challenge when we attempt to describe paramedic practice [18]. A paramedic who works in remote or isolated communities where healthcare resources may be more difficult to access likely requires additional or varied competencies when compared to those whose practice in an urban setting. Similarly, simply adopting the competencies of existing international frameworks [19,20,21] is likely to discount regional differences in healthcare policy priorities and approaches, societal expectations, resources, and access to and distribution of services. Even if some competencies can be generalized, translating descriptions of paramedic practice from one jurisdiction to another ignores this local variation and contextual needs, leaving room for ongoing—or introducing new—alignment concerns.

We previously examined competency framework development processes and outputs in detail [22]. Despite being a necessary link between practice and downstream processes such as curriculum or assessment [23], competency frameworks often fail to adequately define or describe practice, in part because of the methods or conceptual frameworks applied, leading to potential for missed details that threaten the validity and utility of the output. In response, we offered improved development guidance that included a conceptual framework to explore practice informed by systems thinking [23]. Thus, we argue for the importance of accurately identifying and reporting competencies in a way that promotes access to proficiency in these competencies through entry-to-practice education.

In this study, we leverage systems thinking as a conceptual framework intended to make visible features of ‘practice in context’ that may have or continue to be overlooked when developing competency frameworks [23]. Our intention here is not to develop the competency framework but to identify the features that would need attention when that step is taken. While a competency framework can never fully represent the influences, contexts, or competencies required for professional practice, in this paper, we outline and demonstrate our efforts to apply our guidance related to competency framework development to better represent paramedic practice, support the development of the National Competency Framework for Paramedics (NCFP), and demonstrate the utility of that guidance in practice.

## 2. Methods

### 2.1. Overview

We used a systems-thinking approach, which includes both ecological systems theory [24] and complexity theory [25], to guide our work. These two theories were recently combined by Batt et al. [23] as a means of better examining the various levels that may be useful in competency framework development. When examined using this combined systems lens, systems thinking obligates attention to micro, meso, exo, macro, supra-macro, and chronosystem levels (described in more detail below), providing new and often ignored insights. This approach permits the use of multiple methods and assumes an interpretive but predominantly realist philosophical stance [26]. In this study, we used a literature review and system-mapping process to identify relevant features that might be useful in structuring a competency framework. We describe how we defined our target system (a necessary boundary) and details about and positionality of the individuals who participated in the work (i.e., our development group), and then we outline the existing framework and mapping process. We then present findings organized using the system-thinking approach described above.

### 2.2. Defining the System

First, we defined the system of paramedic practice for this study as a system to provide person-centred care in emergency, urgent, unscheduled, and scheduled care contexts by means of educated and regulated paramedics to reduce morbidity and mortality, improve quality of life and experience, and increase access to healthcare. Sensitized by our conceptual framework [23], we identified the relationships between system levels and components in ‘real-world’, non-linear contexts.

### 2.3. Positionality of the Researchers

A development group (DG) led the data collection, analysis, and drafting of the competency framework for the NCFP project. The DG lead (A. Batt) recruited members to represent a variety of clinical practice, education, research, governance, regulatory, policy, leadership, and advocacy experiences within and outside of paramedicine. The DG was composed of 19 individuals from across Canada with experience across multiple contexts of paramedic practice in urban and rural settings, including emergency, community, military, remote and isolated, critical, substance use, special operations, interprofessional primary, and palliative care. They also had experience and expertise in regulation, clinical governance, health systems leadership, research, education, policy and strategy, social work, and private industry. Most of the group identified as Canadian, female, and white, had postgraduate qualifications, and resided in Ontario, followed by British Columbia. Several of the group were bilingual (English and French), while the remainder spoke English as their primary language. P. Mason is a member of the Mohawks of the Bay of Quinte First Nation.

This collaborative project was conducted on colonized Indigenous lands now referred to as Canada. These lands are home to the many diverse First Nations, Inuit, and Métis Peoples whose ancestors have stewarded this land since time immemorial.

### 2.4. Existing Descriptions of Practice

While it can be difficult to identify existing representations of practice, we identified several sources that were used as stimuli for additional sources until we had reached a representative sample of how the community was describing itself. Our explorations of paramedic practice in Canada started with existing descriptions—namely the National Occupational Competency Profile 2011 [14], the Paramedic Profile [27,28], the Canadian Paramedicine Education Guidance document [29], and the Principles to Guide the Future of Paramedicine in Canada [18]. Following this, we sought contemporary literature related to paramedicine in Canada [30,31], informed by two recent comprehensive reviews of the Canadian paramedicine literature [32,33].

### 2.5. System Mapping

We enacted our systems thinking using a multi-stage system-diagramming approach [34]. First, we aimed to capture and represent layers of our complex system using “onion diagrams” [34] (See Appendix A). This also allowed us to brainstorm potential elements and features of the system. Next, we used “rich pictures” [34,35] to explore paramedic practice in differing contexts. Some of these were created by attendees at a community paramedicine workshop in February 2020 after we asked attendees to draw rich pictures of what it is that paramedics do in the community. We then translated these initial ‘messy’ visual explorations of the system into an overall concept map [34,36] intended to organize and represent our developing knowledge (See Appendix A).

Next, we undertook an iterative process of “influence mapping” [34] to identify the relationships between elements in the concept map and as a means of capturing and expressing the complex web of relationships within and between levels (See Appendix A). This allowed us to analyse identified items and their relationships, particularly with how systems thinking is organized and with insights from the paramedicine literature described above. For example, we used the previously published system-level principles intended to guide paramedicine in Canada [18] to identify gaps and/or where alignment could be found. Finally, we combined the results of our diagramming methods (See Appendix A). and insights from the paramedicine literature to create a system map [34] of paramedic practice in Canada, collectively visually representing the complex systems, various components, and their interactions—see Figure 1.

Using iterative group sessions, the systems map was refined at several instances from 2021 to 2024 and left open to identify contemporary data sources that might have emerged as we continued developing the NCFP. Copies of these diagrams are contained in Appendix A. Next, we describe our findings, illustrate the complexity evident at each system level, and offer some potential implications for development of the NCFP.

## 3. Results

### 3.1. Person-Centred—Putting Patients and Their Communities First

Clinical microsystems are embedded in larger systems and are by their definition “patient centred” [23,37]. This obligates attention to the person receiving care and their individual circumstances [18]. When we applied this lens, we were sensitized to the fact that influences on individuals’ health status are often complex [38]. People may have a history of chronic medical conditions, social barriers, or other influences on their health that paramedics should be aware of in order to inform a collaborative approach to decision making, care management, and disposition planning [38,39,40]. These may include multiple, often intersecting, social and structural determinants of health such as class, income, poverty, housing, education, employment status, food security, access to health services, environment, social exclusion, social safety nets, gender, disability, race, ethnicity, and early childhood development [41,42]. Additional influences on peoples’ health may include age, cultural norms, colonial influences, religious beliefs, dependencies, and occupational demands [41,43,44,45]. These findings indicated the importance of social determinants of health, socially responsive advocacy, and the need for culturally competent paramedics.

### 3.2. Microsystem—Considering the Paramedic as a Person

The microsystem refers to the immediate clinical practice environment and all components within it [37]. This necessitates attention to the people involved, their relationships, and their influences on person-centred care [23]. We observed that paramedic practice is unique in that it can encompass all forms of clinical presentations, in unpredictable environments, and with diverse social and cultural contexts [46,47,48,49]. Paramedics are regularly subject to emotionally challenging work, violence [50,51], high rates of occupational injury [48,52,53], stress and mental health issues [54,55,56,57], poor physical health status [18,58,59], and the effects of shift work [60,61,62,63]. In addition, paramedics are often required to make clinical decisions with limited information [64,65,66]. Despite such limited information, they must maintain a high level of situational awareness [67] and must possess other non-clinical attributes such as empathy, honesty, professionalism, conflict resolution, and assertiveness [31,68,69,70]. Our observations of features at this level of the system highlighted the need to engage meaningfully with Indigenous communities around the impacts of colonialism and systemic racism and the implications for practice [71]. A focus on paramedic health and wellbeing should be considered when developing the competency framework. Finally, our findings supported the pursuit of inclusivity, diversity, equity, and accessibility when developing the NCFP [72].

### 3.3. Mesosystem—Appreciating the Influence of People, Policies, and Procedures

The mesosystem represents the interactions that occur between people and the enactment of policies and procedures [23]. While healthcare services exist at the exosystem level (described in more detail below), the delivery of such services takes place via the mesosystem, and this obligates attention to policies, relationships, and professional values. Policy trends at the exosystem level can inform aspects of interprofessional care, and this is enacted in the mesosystem. Within the mesosystem, paramedics interact and communicate with other agents, directly and indirectly related to patient care [52]. These include people receiving care, caregivers and care team members, members of the interdisciplinary team (e.g., other health and social care professionals, personal support workers), public safety personnel (e.g., dispatchers, police, and fire personnel), health services leadership, and members of the public [73,74]. Differences in professional values and cultural expectations could be another instance where practice (reflected in a competency framework) and care needs are poorly aligned. Interprofessional relationships risk being overlooked when professions develop a competency framework, as defining professional competencies may be viewed as an activity exclusive to that profession and may disregard the role of other agents and the relationships between them. Our findings at the mesosystem level suggest that it is necessary to engage other health and social care professionals, people receiving care, caregivers, Indigenous paramedics and communities, and the public when developing the NCFP [75,76].

### 3.4. Exosystem—Identifying Service Delivery Models and Contexts of Paramedic Practice

The exosystem refers to the community level or the service delivery level (e.g., hospitals, clinics, healthcare services) [23]. This requires identifying and attending to different models of service delivery, populations served, and places in which paramedic practice is enacted. We identified multiple forms of paramedic service delivery, which included emergency response, patient transport, community paramedicine, virtual care, and clinic- or hospital-based practice [10,18,38,77,78]. Each of these models of care influence the practice of paramedics within them. For example, community engagement and “practice in context” distinguish community paramedicine models [79], and these along with integration models of paramedic practice have emerged as a solution to local healthcare needs. Many paramedicine models evolved in response to the COVID-19 pandemic, embracing further social initiatives and expanding the diverse communities and populations served by programs [30,77,80,81]. Research has highlighted paramedics providing care for people with non-urgent presentations [82] and palliative needs [46,83]. Paramedic practice is enacted through various models in remote and rural, military, industrial, and expedition, setting [2,84]. Our findings in the exosystem directly suggest the need to explore a variety of diverse practice settings and models of care when developing the competency framework. This might be achieved, for example, by establishing working groups composed of those with expertise in these various settings and models.

### 3.5. Macrosystem—Acknowledging Regional, Provincial, and Federal Influences

The macrosystem contains regional, provincial, and federal influences such as government policies, culture, religious movements, the economy, and societal issues [23]. This requires attention to how paramedics are educated, regulated, funded, and positioned to provide care. We identified that paramedics are governed by a variety of clinical oversight or regulatory models across Canada, along a spectrum from not regulated, to government body regulated, to self-regulated [85,86]. Regulatory models exert significant influence on education, service design and delivery, clinical scope of practice, and clinician autonomy. Paramedic education standards (which are often provincially dictated) are as diverse as the regulatory models that exist and vary significantly in duration, form, and intended outcomes across, and within, jurisdictions. Education exerts subsequent influence on clinician autonomy, career development, decision making, and the development of both clinical and non-clinical attributes. Socio-political impacts on health are also evident at this level in relation to health policy choices such as promoting (or not) public health interventions, funding (or de-funding) services, expanding (or limiting) access to certain health services, and the introduction of legislation that impacts health. The funding of paramedic services ranges from government funded at all levels to subsidized programs and those that depend on payment from individuals or their health insurance plans [87]. Our observations at this level suggest the need to attend to the variations in regulation, education, funding, scope of practice, and other elements across Canada when developing the NCFP. This may be achievable by avoiding granularity when identifying competencies that may be influenced by such variations and prioritizing a focus on common features. Broad representation and the use of consensus-based approaches will be required to ensure the competency framework is applicable across Canada.

### 3.6. Supra-Macrosystem—Harnessing Insights from Global Events to Determine Influences on Practice

Global influences such as pandemics, geopolitical issues such as war, and mass immigration exist at the supra-macro level [23]. Although such issues may seem far removed from the microsystem of paramedic practice [88], they can exert significant influence and contribute to the complexity within the system. For example, although paramedics may be some distance from the focal point of the war, an influx of refugee populations to their local area will influence their clinical practice on several levels. The unpredictable nature of war, and the subsequent impact migration may have on practice, illustrates the challenge of determining in advance the competencies required of health professionals but highlights the need for improved guidance by which to approach such uncertainty. Our observations at this level suggest the need to explore and attend to emerging concepts, evolving events [33], and societal shifts and a need to ‘future-proof’ competencies related to these issues, while acknowledging the inability to ‘predict’ future competency needs. This can be mitigated by regular updates and maintenance of the competency framework to ensure it remains contemporary and relevant.

### 3.7. Chronosystem—Reflecting on History to Better Prepare for the Future

The chronosystem refers to the changes within the system and to the dynamic change to the totality of the system over time [23]. Paramedic practice has evolved since 2011 to provide increasingly complex levels of autonomous care in highly variable clinical contexts. In addition, paramedic education has progressed, and service delivery models have evolved across Canada [18]. A broader perspective at this level would suggest that changes have occurred over time in response to changing societal expectations of health and social care and changing public healthcare policy. Modern-day health service perspectives include a move toward increased equity of access to healthcare (resulting in changes to how paramedics deliver care) and reflecting on demographic changes within society and within professions (affecting the clinical workload and the impact on paramedics). Widespread societal influences such as the Truth and Reconciliation Commission’s (TRC) Calls to Action [76] and cultural influences such as the increasing popularity of the internet and social media (which have influenced how paramedics are educated and should behave) must also be acknowledged. Our observations on how the system has changed over time suggest that competency framework development must attend to equity, the impacts of increasing workloads, and the TRC’s Calls to Action related to health.

## 4. Discussion

Competency frameworks play an integral role in reflecting and shaping the practices of the health professions they are intended to represent. They provide a structured way for the profession, public, educators and accreditors, regulators, and policy makers to guide strategies and ways forward, assuming they are trustworthy in their representation of practice and the profession. Using recently developed guidance that provides a conceptual map for investigation, we examined paramedicine comprehensively, such that findings could inform future paramedic competency framework development. In summary, we identified several interacting layers and influences centred around patients, as well as their unique community, social, and environmental contexts. Bringing these influences and insights together requires a structured approach and a coordinated team to consider these insights in more detail and how best to capture and translate them for inclusion when developing the NCFP. This attempt to better reflect paramedic practice used a recently published systems-thinking approach to identify the features of professional practice [23]. This led to influences that had previously been ignored. From this experience, we offer three critical reflections on implementing such an approach.

First, we enacted a systems-thinking approach within a six-step competency framework development model [89]. The steps of this model include (1) identifying purpose, intended uses, scope, and stakeholders; (2) theoretically informed ways of identifying the contexts of complex, “real-world” professional practice, which includes (3) aligned methods and means by which practice can be explored; (4) the identification and specification of competencies required for professional practice; (5) how to report the process and outputs of identifying such competencies; and (6) built-in strategies to continuously evaluate, update, and maintain competency framework development processes and outputs. Identifying the features and elements of the system of paramedic practice is part of Step 2 of this model and is addressed in this study. However, our systems-thinking approach was influenced by decisions made during Step 1 of the six-step model. For example, the scope of the competency framework (e.g., to not include Emergency Medical Responders) placed some natural boundaries on the ‘scope’ of the system and the elements that required identification (e.g., the NCFP did not address leadership competencies). On reflection, we considered the boundaries placed by Step 1 of the NCFP development process to be overall beneficial in that they helped to shape the structure of the system that we needed to understand. While such boundaries may be helpful in focusing a systems-thinking approach, they also represent a risk in failing to identify elements that are considered ‘out of scope’. Developers should remain aware of these implications when choosing to develop competency frameworks using either or both guiding models.

Second, the need to consider geographical, discipline-specific, societal, social, and cultural boundaries is a challenging aspect of any process. It is inevitable that we missed some features of the system with our approach, and yet it is also likely that we included too many other features to make the model useful or precise. We created a model of paramedic practice, and as with all models, it is a partial representation that is socially constructed. However, we made concerted efforts to mitigate the impacts of ‘missing features’, including broadly engaging with the literature on paramedicine and health systems, recruiting a development group with diverse identities and experiences, and engaging a systematic approach to consider each system level. In doing so, we also had to mitigate accounting for everything in the system, resulting in an unusable model. We achieved this by guidance from the scope of the overall project and by keeping our focus on paramedic practice through intimate involvement of those in practice. We suggest developers of competency frameworks pay particular attention to engaging a group of experts from within and outside of the profession, select the members of the group to provide a diversity of intersecting positions and experiences, and involve them in enacting the process of identifying (and subsequently exploring) the features of the system.

Finally, given the pan-Canadian national scope of the NCFP, it was deemed necessary to consider the macro and supra-macrosystems in our understanding of the system of paramedic practice. Given the dynamic nature of systems at this scale, we revisited our understanding of the system as events unfolded to ensure we had attended to any novel considerations. Examples of such events included civil unrest in Canada related to COVID-19 vaccine mandates [90], the need to protect the reproductive rights of women given developments in the USA [91], ongoing economic and financial pressures [92], wildfires across the country [93], and the outbreak of conflict in Ukraine [94]. These and other events had implications for all health professions’ practice including the safety and wellbeing of professionals, cultural competency expectations, awareness of health and social system access, and the importance of social and structural determinants of health. The need to revisit and iterate an understanding of the system should be considered from the outset. How best to translate these events for the purpose of a competency framework remains a challenge but is worthy of pursuit. A process by which to systematically undertake this ongoing element would be beneficial and should be considered by those developing competency frameworks.

## 5. Translating and Transforming These Findings

The collection of these findings will require the work of translation and transformation into a competency framework, which we will now develop. This is an iterative process guided by various factors, influences, and their interactions identified in this study. This process is expected to encounter, navigate, and resolve issues of messiness, comprehensiveness, and of potential feasibility and utility. However, we expect that this competency framework development work will be strengthened by the comprehensive foundation presented here in this paper. We recommend further research into the use of this systems-thinking approach to describe practice, not just for developing competency frameworks but for better understanding of health professions practice in context. In addition, future research should seek to determine methods by which uncertainty (related to, for example, decision making, economic impacts, outcomes) can be accounted for and methods by which evolving events in the macrosystem and supra-macrosystem can be identified, considered, and reflected.

## 6. Conclusions

Paramedic practice in Canada continues to evolve and there is a duty to ensure that documents that guide education, assessment, regulation, and professional development reflect the evolving complexity of contemporary paramedic practice and outline the features required for competent paramedic practice across Canada. In this study, by leveraging a conceptual framework informed by systems thinking, we identified the diverse contexts in which paramedic practice is enacted in Canada. These findings suggest that such contexts be considered and integrated when attempts are made to describe or represent paramedic practice through competency frameworks. Not doing so presents a risk to our understanding of professional practice and ultimately the validity and utility of proposed frameworks. Doing so will help us to describe how paramedic practice comprises interdependent health and social care aspects. This, in turn, suggests that as a profession we have an opportunity (through a robust competency framework) to better align education with practice, to prepare learners for their professional roles, to align with system advances, and to improve service delivery models to meet the needs of the communities they serve.

## Figures and Tables

**Figure 1 healthcare-12-00946-f001:**
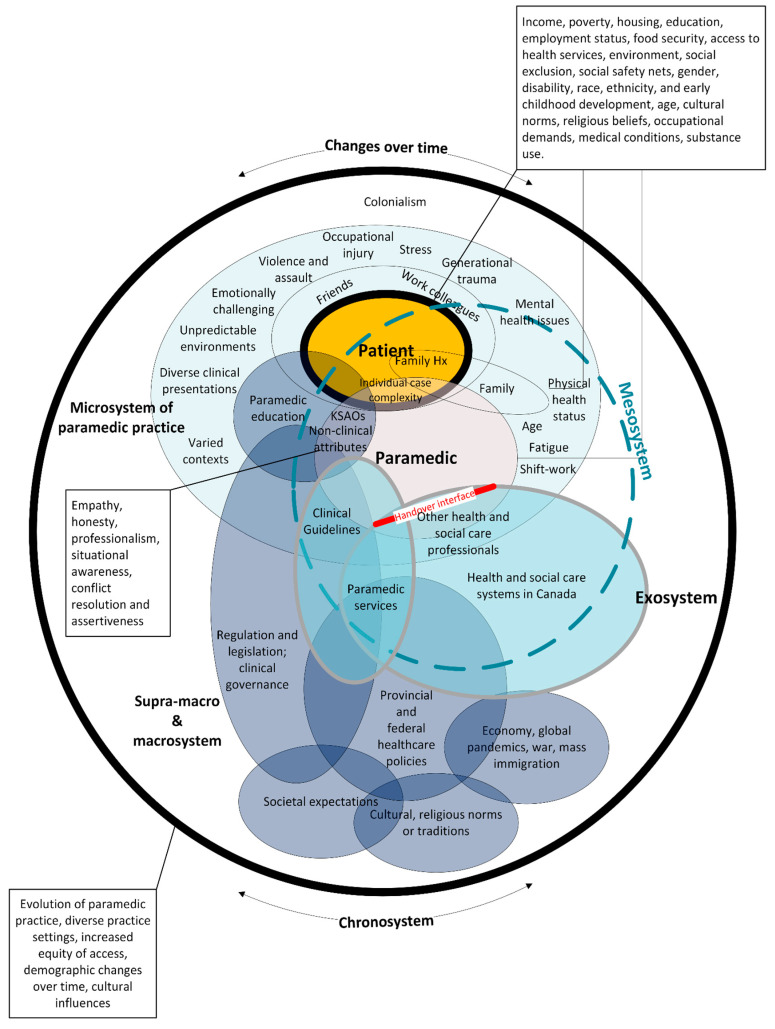
Systems map of paramedic practice in Canada. This figure illustrates the proposed relationships or interactions between various system levels and was created via a process of influence mapping, whereby the relationships between various features of the system were identified. This figure places the person receiving care as the central focus of the system of paramedic practice in Canada. There are multiple areas of influence between system levels (and their features). The mesosystem is ‘constructed’ via interactions between the microsystem and the exosystem. The chronosystem represents both changes in system features over time and the dynamic change to the totality of the system over time. Notes: the size of elements is irrelevant; overlaps do not illustrate significance but rather illustrate influence; and the model is a partial representation of paramedic practice in Canada and does not claim to be a validated or complete map of the system.

## Data Availability

Data are available from the corresponding author on reasonable request.

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
