# Peer review of "Identifying Features of a System of Practice to Inform a Contemporary Competency Framework for Paramedics in Canada"

_healthcare, 2024, doi:10.3390/healthcare12090946_

Round 1

Reviewer 1 Report

Comments and Suggestions for Authors

You must check the spelling of the text.

The bibliography must be reviewed to follow the editorial standards of
this diary.

It is advisable to delve into some aspects of the discussion of the article that are not entirely clear and that must be brought up so that the reader is clear about the difficulties that the researchers have presented, as well as the possible studies to be carried out in the future. The authors should clarify this issue further.

It is advisable to delve deeper into the conclusions of the article. Some of the contributions are not clear. It is advisable to reread and meditate after reading the results of this research work.

Comments on the Quality of English Language

You must check the spelling of the text.

The bibliography must be reviewed to follow the editorial standards of
this diary.

It is advisable to delve into some aspects of the discussion of the article that are not entirely clear and that must be brought up so that the reader is clear about the difficulties that the researchers have presented, as well as the possible studies to be carried out in the future. The authors should clarify this issue further.

It is advisable to delve deeper into the conclusions of the article. Some of the contributions are not clear. It is advisable to reread and meditate after reading the results of this research work.

Reviewer 2 Report

Comments and Suggestions for Authors

Thanks for submitting the article. Please find the following comments:

- for the abstract, add ':' after Introduction; Methods, Findings, Conclusion

- pg. 2, line 90, nationally in Canada as reader may ask if you are referring to Canada

- pg. 2, line 99, Regional variations if more than one perspectives

- pg. 3, line 121-122, not consistency with abstract as it was stated to revise the competency framework in the abstract

- pg. 4, first paragraph, put full form for AB and PM as reader from other countries may not understand

- pg. 4,  line 167-169 need citiation

- pg.6, line 286, which organization is responsible for making the paramedic education standards? Pls quote

Comments on the Quality of English Language

Good. Minor point to be amended as mentioned in the 'Comments section above'

Reviewer 3 Report

Comments and Suggestions for Authors

REVIEW REPORT FOR THE STUDY “IDENTIFYING FEATURES OF A SYSTEM OF PRACTICE TO INFORM A CONTEMPORARY COMPETENCY FRAMEWORK FOR PARAMEDICS IN CANADA”
Journal: Healthcare
The paper "Identifying Features of a System of Practice to Inform a Contemporary Competency Framework for Paramedics in Canada", performs an analysis to better identify, explore, and represent professional practice when revising a national competency framework for paramedics in Canada.
Title and summary. The title and abstract express well the object of study, objectives, and results of the article.
Structure of the article. The contents are well organized and they adhere to the IMRaD structure. It includes a theoretical framework of the research problem but at this point, I suggest the authors incorporate some other bibliographic references that I miss in the text:
Ron R Bowles; Catherina van Beek; Gregory S Anderson. Justice Institute of British Columbia, Canada Four dimensions of paramedic practice in Canada. Australasian Journal of Paramedicine: 2017;14(3).
O'Meara P, Stirling C, Ruest M, Martin A. Community paramedicine model of care: an observational, ethnographic case study. BMC Health Serv Res. 2016 Feb 2;16:39. doi: 10.1186/s12913-016-1282-0. PMID: 26842850; PMCID: PMC4739332.
Focusing on the opportunity of the study, it must be said that it is useful work since it covers one of the major problems resulting from a health care system.
Materials and methods.
Regarding the material and methods section, the methodology is tailored to the object of study and the objectives and is explained in a transparent manner while it has been validly applied to guarantee the results.
Nevertheless, It is suggested, for research improvement, to address a proposal of a mathematical modeling for decision making under uncertainty in the Bronfenbrenner ecological model.
Results.
The results are significant and they are presented in an adequate and understandable way not only through narration but also with self-explained tables and figures that are also well elaborated in terms of presentation. The results justify and relate to the objectives and methods and the results are of sufficient interest.
It is suggested, for better understanding by the readers, to include graphics that explain and detail: the Microsystems that influence the paramedic framework; the Variables involved in the different contexts that form the paramedic microsystem; the Relationships that are established in the mesosystem of the paramedic framework, the Exosystem that surrounds the paramedic framework. and the Macrosystem in which the paramedic framework is involved.
Discussion.
The discussion appropriately compares the study results with other works, highlighting the main study findings. However, I would propose the inclusion of two bibliographic references in the discussion section:
Eaton G, Tierney S, Wong G, et al. Understanding the roles and work of paramedics in primary care: a national cross-sectional survey. BMJ Open 2022;12:e067476. doi:10.1136/ bmjopen-2022-067476.
Batt A, Williams B, Rich J and Tavares W (2021) A Six-Step Model for Developing Competency Frameworks in the Healthcare Professions. Front. Med. 8:789828. doi: 10.3389/fmed.2021.789828.
It is also suggested to include in the discussion the comparison of Bronfenbrenner's model with other developmental theories such as Jean Piaget's cognitive development theory, the principles of Gestalt psychology, Carl Jung's analytical psychology, Erik Erikson's psychosocial stages, or attachment theories proposed by John Bowlby and Mary Ainsworth.
Bibliography.
The 56.98% of the bibliography cited in the study belongs to the previous five years.
The authors should complete bibliographical reference number 83.
Overall, this is an interesting study and should be considered for publication in Healthcare, once the main proposed revisions have been resolved.
